# A Proteomic Approach Identified TFEB as a Key Player in the Protective Action of Novel CB2R Bitopic Ligand FD22a against the Deleterious Effects Induced by β-Amyloid in Glial Cells

**DOI:** 10.3390/cells13100875

**Published:** 2024-05-19

**Authors:** Beatrice Polini, Lorenzo Zallocco, Francesca Gado, Rebecca Ferrisi, Caterina Ricardi, Mariachiara Zuccarini, Vittoria Carnicelli, Clementina Manera, Maurizio Ronci, Antonio Lucacchini, Riccardo Zucchi, Laura Giusti, Grazia Chiellini

**Affiliations:** 1Department of Pathology, University of Pisa, 56100 Pisa, Italy; beatrice.polini@farm.unipi.it (B.P.); c.ricardi@student.unisi.it (C.R.); vittoria.carnicelli@unipi.it (V.C.); riccardo.zucchi@unipi.it (R.Z.); 2Department of Translational Research and New Technologies in Medicine and Surgery, University of Pisa, 56126 Pisa, Italy; l.zallocco@gmail.com; 3Department of Pharmaceutical Sciences, University of Milan, 20133 Milano, Italy; francesca.gado@unimi.it (F.G.); rebecca.ferrisi@unimi.it (R.F.); 4Department of Pharmacy, University of Pisa, 56126 Pisa, Italy; clementina.manera@unipi.it; 5Department of Medical, Oral and Biotechnological Sciences, University G. D’Annunzio of Chieti-Pescara, 66100 Chieti, Italy; mariachiara.zuccarini@unich.it (M.Z.); maurizio.ronci@unich.it (M.R.); 6Interuniversitary Consortium for Engineering and Medicine (COIIM), 86100 Campobasso, Italy; 7Department of Clinical and Experimental Medicine, University of Pisa, 56126 Pisa, Italy; antonio.lucacchini@unipi.it; 8School of Pharmacy, University of Camerino, 62032 Camerino, Italy

**Keywords:** cannabinoid receptor type II (CBR2), autophagy, neuroinflammation, proteomic, β-amyloid, TFEB

## Abstract

Neurodegenerative diseases (NDDs) are progressive multifactorial disorders of the nervous system sharing common pathogenic features, including intracellular misfolded protein aggregation, mitochondrial deficit, and inflammation. Taking into consideration the multifaceted nature of NDDs, development of multitarget-directed ligands (MTDLs) has evolved as an attractive therapeutic strategy. Compounds that target the cannabinoid receptor type II (CB2R) are rapidly emerging as novel effective MTDLs against common NDDs, such as Alzheimer’s disease (AD). We recently developed the first CB2R bitopic/dualsteric ligand, namely FD22a, which revealed the ability to induce neuroprotection with fewer side effects. To explore the potential of FD22a as a multitarget drug for the treatment of NDDs, we investigated here its ability to prevent the toxic effect of β-amyloid (Aβ_25–35_ peptide) on human cellular models of neurodegeneration, such as microglia (HMC3) and glioblastoma (U87-MG) cell lines. Our results displayed that FD22a efficiently prevented Aβ_25–35_ cytotoxic and proinflammatory effects in both cell lines and counteracted β-amyloid-induced depression of autophagy in U87-MG cells. Notably, a quantitative proteomic analysis of U87-MG cells revealed that FD22a was able to potently stimulate the autophagy–lysosomal pathway (ALP) by activating its master transcriptional regulator TFEB, ultimately increasing the potential of this novel CB2R bitopic/dualsteric ligand as a multitarget drug for the treatment of NDDs.

## 1. Introduction

The transcription factor EB (TFEB) is a central regulator of the autophagy–lysosomal pathway (ALP) [1,2], which is a major mechanism for degrading intracellular macromolecules, including long-lived proteins, aggregated misfolded proteins, and abnormal cytoplasmic organelles, and maintaining cellular homeostasis (Figure 1).

It is widely recognized that dysfunction in the ALP is a pathogenic feature shared by multiple adult-onset neurodegenerative disorders (NDDs), including Alzheimer’s, Parkinson’s, and Huntington’s diseases [3,4,5]. In general, the accumulation of intracellular aggregates in the brain, which leads to synaptic dysfunction and ultimately neuronal death, is a common feature for all these pathologies.

Despite extensive research efforts, the complex etiology of these diseases is not yet clear, and limited treatment options are currently available.

TFEB is widely expressed in the CNS, including in neurons and astrocytes [6,7,8]. In physiological conditions, TFEB localizes to the cytosol and rests on the lysosomal surface, where upstream kinases, such as rapamycin complex 1 (mTORC1) [9], can phosphorylate it. Thus, the inhibition of mTOR, induced by starvation and lysosomal stress, promotes TFEB dephosphorylation and its nuclear translocation [1,10]. Nuclear TFEB increases the transcription of genes involved in the regulation of lysosomal, autophagic, and retromer function, collectively called the coordinated lysosomal expression and regulation (CLEAR) network. Notably, nuclear TFEB localization is decreased in different NDDs in which protein aggregation takes place [6,8,11,12], thus suggesting that defective nuclear translocation of TFEB is related to impaired protein homeostasis in neurons. TFEB overexpression is widely known to increase the number of autophagosomes, to promote the generation of new lysosomes, and to increase the autophagic flux [1,13,14,15]. Therefore, inducing intracellular clearance through the induction of TFEB activity may represent an appealing therapeutic intervention for the treatment of NDDs.

In recent years, considerable advances in cannabinoid research have renewed interest in targeting components of the endocannabinoid system (ECS) as treatment options in central nervous system (CNS) disorders and NDDs [16,17]. The ECS is a complex molecular system that plays critical roles in multiple physiological processes such as homeostasis, neurogenesis, neuroprotection, and inflammation [18,19]. Elements of the ECS comprise the endogenous ligands, endocannabinoids (eCBs) anandamide (AEA) and 2-arachidonoylglycerol (2-AG), their synthesizing and metabolizing enzymes, and the cannabinoid receptors type 1 (CB1R), type 2 (CB2R), and other putative cannabinoid receptor candidates [20,21] (Figure 2).

While most research has investigated CB1R, which is highly expressed in nearly all brain regions, CB2R in the brain has started to attract considerable interest only in recent years, having been considered for a long time exclusively a peripheral-type receptor [22]. During the last two decades numerous experimental studies have provided robust evidence that CB2R seems to be involved in the modulation of different neurological disorders characterized by neuroinflammatory processes and microglial cell activation [23], suggesting the therapeutic potential of natural and synthetic CB2R ligands in the treatment of neurodegenerative proteinopathies, such as Alzheimer’s and Parkinson’s diseases [24,25].

Notably, a very recent report highlighted a new mechanism allowing CB2R to regulate autophagy (ATG), lipid metabolism, and inflammation in an animal model of postoperative cognitive dysfunction (POCD) through the modulation of TFEB activity [26], further increasing the potential of CB2R targeting for therapeutic intervention in NDDs.

Our group has recently developed the first CB2R bitopic/dualsteric ligand, namely FD22a, which has been shown to display beneficial biological responses both in vitro and in vivo with fewer side effects [27].

Bitopic/dualsteric ligands, which are hybrid compounds composed of orthosteric and allosteric pharmacophoric units, represent one of the most promising strategies of targeting G protein-coupled receptors (GPCRs) [28]. Indeed, this approach allows the exploitation of favorable characteristics of the orthosteric and the allosteric site by a single ligand molecule, including an increased affinity or selectivity for the target receptor, often associated with functional selectivity (i.e., bias signaling pathway activation), reduced off-target activity, and therapeutic resistance [29].

Notably, novel CB2R bitopic ligand FD22a met requirements typical of the bitopic ligand, such as receptor–subtype selectivity and biased signaling for cAMP inhibition versus βarrestin2 recruitment, while revealing significant neuroprotective effects, such as the ability to efficiently combat the inflammatory process in human microglial cells and to display antinociceptive activity in vivo [27]. In addition, computational studies clarified the binding mode of this compound inside the CB2R, further confirming its bitopic nature [27].

To explore in more detail the potential of newly developed CB2R bitopic ligand FD22a to target neurodegeneration, in the present study we investigated the ability of FD22a to counteract the detrimental effects produced by the neurotoxic Aβ fragment 25–35 (Aβ_25–35_) [30] in human cellular models of neurodegeneration, such as human microglial (HMC3) and human glioblastoma–astrocytoma (U87-MG) cell lines.

In these models FD22a prevented the cytotoxic and proinflammatory effects of Aβ_25–35_ and efficiently counteracted the depression of autophagy caused by Aβ_25–35_.

Moreover, protein expression profiling, combined with pathways analyses, revealed that FD22a was able to potently stimulate the ALP pathway through TFEB activation, in turn reversing Aβ_25–35_ neurotoxicity by promoting intracellular clearance.

## 2. Materials and Methods

### 2.1. Cell Cultures, Reagents, and Treatments

Human glioblastoma (U87-MG, ATCC^®^ HTB-14™, Manassas, VA, USA) and human microglial clone 3 (HMC3, ATCC^®^ CRL-3304™, Manassas, VA, USA) cell lines were cultured in high-glucose DMEM supplemented with 10% fetal bovine serum and a 1:1 antibiotic mixture of streptomycin (100 g/mL) and penicillin (100 U/mL) (Sigma-Aldrich, Milan, Italy) at 37 °C in 5% CO_2_ humidified air.

Aβ_25–35_ peptide (NH_2_-Gly-Ser-Asn-Lys-Gly-Ala-Ile-Ile-Gly-Leu-Met-COOH) (A4559, Sigma-Aldrich, Milan, Italy) was initially dissolved in double-distilled water to obtain 1 mM concentration and stored at −20 °C. To form aggregated diffusible oligomers, the solution was incubated at 37 °C for 5 days [31], then diluted in medium to the indicated concentration, just prior to cell treatments. FD22a was dissolved in DMSO to obtain a 50 mM stock solution which was kept at 4 °C. Before the experiments FD22a stock solution was diluted into the cell culture medium to the desired experimental concentration, and the final DMSO concentration was maintained no higher than 0.1%. Vehicle-treated cells (0.1% DMSO) were used as control (Ctrl).

In all the experiments 24 h after seeding, cells were exposed to pretreatment with FD22a for 24 h and then exposed to Aβ_25–35_ at the pertinent concentration (10 µM for U87-MG and 1 µM for HMC3). After 48 h, cells were processed according to the specific experiment protocol.

#### 2.1.1. MTT (Cell Viability Assay)

The 3-(4,5-dimethylthiazol-2-yl)-2,5-diphenyltetrazolium bromide (MTT, Sigma-Aldrich, Milan, Italy) reagent was used to test the effect of FD22a on cell viability. Briefly, after treatment, cells were incubated with MTT (0.5 mg/mL) for 4 h at 37 °C. The formazan products were dissolved in DMSO. An automated microplate reader (BIO-TEK, Winooski, VT, United States) was used to quantify absorbance at 540 nm. Cell viability was expressed as the percentage of control cells.

#### 2.1.2. Release of Inflammatory Cytokines

Concentrations of proinflammatory interleukin 6 (IL-6), tumor necrosis factor α (TNFα), and anti-inflammatory interleukin 10 (IL-10) were quantified by specific ELISAs (RAB0306 (IL-6), RAB0476 (TNFα), and RAB0244 (IL-10), Sigma-Aldrich, Milan, Italy). After pertinent treatment, culture media were collected and stored at −80 °C until analysis.

#### 2.1.3. Gene Expression Analysis

Total RNA was extracted using the RNeasy Mini kit (74104, Qiagen, Hilden, Germany) and a Qubit v.1 fluorometer plus Qubit RNA HS Assay Kit (Thermo Fisher Scientific, Wilmington, DE, USA) was used to extract and quantify total RNA, on the basis of manual protocol indications.

Extracted RNA (1 μg) was retrotranscribed by using the iScriptTM gDNA Clear cDNA Synthesis Kit (Bio-Rad, Milan, Italy) according to the manufacturer’s instructions, and the obtained cDNA samples were quantified by real-time PCR using a SYBR Green probe and CFX Connect Real-Time PCR Detection System (Bio-Rad, Milan, Italy). The PCR cycle program consisted of an initial denaturation at 95 °C for 30 s followed by 40 cycles of 5 s of denaturation at 95 °C and 15 s of annealing/extension at 60 °C. To verify amplicon specificity and potential primer dimer formation, a final melting protocol with ramping from 65 °C to 95 °C with 0.5 °C increments of 5 s was performed.

Primer sequences (Table 1) were designed by using Beacon Designer Software v.8.0 (Premier Biosoft International, Palo Alto, CA, USA) with a junction primer strategy, whenever possible. To exclude genomic DNA contamination, a negative retrotranscription control was used. The endogenous reference gene GAPDH was quantified for each sample.

All reactions were performed in triplicate and the amount of mRNA was calculated by the comparative critical threshold (CT) method.

### 2.2. Proteomic Analysis

For proteomic analysis, U87-MG human glioblastoma cells were exposed to pretreatment with FD22a (1 μM) for 24 h before being exposed for 48 h with Aβ_25–35_ (10 μM) as described above. After treatment, cells were rinsed with ice-cold PBS and lysed in rehydration solution (7 M urea, 2 M thiourea, 4% CHAPS, 60 mM dithiothreitol (DTT), 0.002% bromophenol blue) added with 50 mM NaF, 2 mM Na_3_VO_4_, 1 μL/10^6^ cells, and protease cocktail inhibitors (Sigma-Aldrich, St. Louis, MO, USA). After stirring and sonication, cells were allowed to rehydrate for 1 h at room temperature (RT) and, thereafter, the solution was centrifuged at 16,000× *g* for 10 min at RT [32]. Protein contents of resulting protein extracts were measured with the Pierce Protein Assay (Thermo Fisher Scientific, Waltham, MA, USA) and bovine serum albumin was used as standard.

Two-dimensional electrophoresis (2DE) was carried out as previously described [33]. Briefly, one hundred fifty micrograms of proteins was loaded on Serva IPG blue strips (SERVA-German Headquarter, Heidelberg, Germany) with a linear pH 3–10 gradient. The second dimension (SDS-PAGE) was carried out by transferring the proteins to 12% polyacrylamide gels. The gels were stained with Ruthenium II tris (bathophenanthroline disulfonate) tetrasodium salt (Cyanagen Srl, Bologna, Italy) (RuBP) [34] and images were acquired by ImageQuant LAS4010 (GE Health Care, Uppsala, Sweden). The analysis of images was performed using Same Spot (v4.1, TotalLab; Newcastle Upon Tyne, UK) software [35]. The spot volume ratios among the four different conditions (control, FD22a, Aβ_25–35_, FD22a + Aβ_25–35_) were calculated using the average spot normalized volume of the six biological replicates. The software included statistical analysis calculations.

#### 2.2.1. In Gel Digestion and Mass Spectrometry

The gel pieces were digested as reported by Giusti et al. 2018 [36]. Peptide MS spectra were recorded manually on the AutoFlex Speed MALDI-TOF/TOF spectrometer (Bruker Daltonics, Leipzig, Germany) operated in positive reflectron mode [37]. Samples unidentified by MALDI-TOF/TOF were analyzed by LC-MS/MS using an UltiMate3000 RSLCnano chromatographic system coupled to an Orbitrap Fusion Tribrid mass spectrometer (Thermo Fisher Scientific, Waltham, MA, USA), operating in positive ionization mode, equipped with a nanoESI source (EASY-Spray NG). Peptides were loaded on a PepMap100 C18 precolumn cartridge (5 µm particle size, 100 Å pore size, 300 µm i.d. × 5 mm length, Thermo Fisher Scientific, Waltham, MA, USA) and subsequently separated on an EASY-Spray PepMap RSLC C18 column (2 µm particle size, 100 Å pore size, 75 µm i.d. × 15 cm length, Thermo Fisher Scientific, Waltham, MA, USA) at a flow rate of 300 nL/min and a temperature of 40 °C, using 0.1% FA in water (eluent A) and 99.9% ACN, 0.1% FA (eluent B). The chromatographic separation was achieved by a two-step linear gradient from 5% to 30% eluent B in 40 min, and from 30% to 55% in 5 min followed by an increase to 90% in one minute, for a total runtime of 56 min.

Precursor (MS1) survey scans were recorded in the Orbitrap, at resolving powers of 240 K (at *m*/*z* 200). Data-dependent MS/MS (MS2) analysis was performed in top-speed mode with a 3 s cycle time, during which most abundant multiple-charged (2+–5+) precursor ions detected within the range of 375–1500 *m*/*z* were selected for HCD activation in order of abundance and detected in the ion trap at a rapid scan rate after fragmentation using 30% normalized collision energy. Quadrupole isolation with a 1.6 *m*/*z* isolation window was used, and dynamic exclusion was enabled for 60 s after a single scan. Automatic gain control targets and maximum injection times were standard and auto for MS1 and 150% and 70 for MS2. For MS2, the signal intensity threshold was 5.0 × 10^3^, and the option “Injection Ions for All Available Parallelizable Time” was set.

Raw data were directly loaded in PEAKS Studio Xpro software 11 (Bioinformatic Solutions Inc., Waterloo, ON, Canada) using the “correct precursor only” option. The mass lists were searched against the UniProt/SwissProt database (downloaded January 2022) restricted to Mammalia taxonomy to which a list of common contaminants was appended (67,666 searched entries). Non-specific cleavage was allowed to one end of the peptides, with a maximum of 2 missed cleavages and 2 variable PTMs per peptide. Additionally, 10 ppm and 0.5 Da were set as the highest error mass tolerances for precursors and fragments, respectively. A −10lg*p* threshold for PSMs was manually set to 35.

#### 2.2.2. Bioinformatic Analysis

To determine the predominant canonical pathways and interaction network involved, differentially expressed proteins obtained from the comparison of FD22a + Aβ_25–35_ vs. Aβ_25–35_ were functionally analyzed using QIAGEN’s Ingenuity Pathway Analysis (IPA, QIAGEN Redwood City, CA, USA, www.qiagen.com/ingenuity, Build version: 321501M, Content version: 21249400, accessed on 15 January 2021). Comparison of the different analyses was performed and potential regulators and downstream functions were investigated as previously described [33].

#### 2.2.3. Western Blot Analysis

Equal amounts of proteins from U87-MG cell lysates (30 μg of proteins) were mixed with Laemmli solution, run in 4–15% polyacrylamide gels (Mini-PROTEAN^®^ Precast Gels, Biorad, Hercules, CA, USA) using a mini-Protean Tetracell (Biorad, Hercules, CA, USA), and transferred onto nitrocellulose membranes (0.2 μm) using a Trans-Blot Turbo transfer system (Biorad) as previously described [33]. Immediately after, WB membranes were stained with 1 mM RuBP and total protein images acquired by ImageQuant LAS4010 (GE Health Care, Uppsala, Sweden). Subsequently, membranes were incubated with primary antibodies (dilution 1:1000). The following antibodies were purchased from Cell Signaling Technology, Beverly, MA, USA: TFEB (D207D) rabbit mAb (#37785), phosphorylated TFEB (E9S8N) rabbit mAb (pTFEBS211; #37681), mTOR (7C10) rabbit mAb (#2983), phosphorylated mTOR (D9C2) XP rabbit mAb (p-mTORS2448; #5536). Immunoblots were developed using the enhanced chemiluminescence (ECL) detection system. The chemiluminescent images were acquired using LAS4010 (GE Health Care Europe, Upsala, Sweden). Semiquantitative analysis of specific immunolabeled bands was performed using ImageQuant TL 7 software.

#### 2.2.4. Statistical Analysis

For cell viability, ELISAs and gene expression analysis results are expressed as the mean ± standard error of the mean (SEM). Statistical analyses were performed using commercial software (GraphPad Prism, San Diego, CA, USA) using ordinary one-way ANOVA followed by Dunnett’s or Tukey’s post hoc tests. Differences for which *p* < 0.05 were considered significant.

For proteomic studies, all analyses were performed at least in triplicate and values were expressed as mean ± standard error (SD). In 2DE experiments, a comparison among the different treatments was performed. The significance of the differences of normalized volume for each spot was calculated by the software Same Spot (v4.1, TotalLab, Newcastle Upon Tyne, UK) including the analysis of variance (ANOVA test). The protein spots that showed significant differences in expression were cut out from the gel and identified by mass spectrometry analyses. The immunoreactive bands obtained in WB experiments were analyzed using ImageQuant TL (GE Health Care). The antigen-specific bands and the total proteins after RuBP staining were quantified. The volume of each band was normalized on total proteins obtained from RuBP staining. The results were expressed as a ratio of optical density. For phosphoproteins a consecutive normalization on the expression level of corresponding proteins was performed. An unpaired *t*-test was used to compare differences among treatments (Prism 7; GraphPad Software, San Diego, CA, USA). Differences for which *p* < 0.05 were considered significant.

## 3. Results

### 3.1. FD22a Prevents Aβ_25–35_-Induced Cytotoxicity in HMC3 and U87-MG Cells

To investigate the potential protective activity of FD22a against Aβ_25–35_-induced cytotoxicity, two different human cell lines, namely human microglial (HMC3) and glioblastoma (U87-MG) cells, were exposed to pretreatment (24 h) with increasing concentrations (0.1–10 μM) of FD22a before being treated for 48 h with 1 or 10 μM Aβ_25–35_, respectively (Figure 3). These two β-amyloid concentrations have been chosen as they reduce cell viability by approximately 50% compared to either HMC3 or U87-MG control cells (Figure 3C,D). Moreover, comparable Aβ_25–35_ concentrations have been previously used by Polini et al. in HMC3 cell culture experiments [31]. Of note, in HMC3 cells no tested concentration of FD22a affected cell viability (Figure 3A), whereas in U87-MG cells, when FD22a was used at a concentration higher than 1 µM, a slight cytotoxic effect was observed (Figure 3B). Vehicle-treated cells (Ctrl) did not show any difference compared to untreated cells (NT).

Cell viability was analyzed by MTT assay. As expected, Aβ_25–35_ significantly reduced cell viability with respect to control cells (Figure 3C,D). When used at 0.1 μM, FD22a did not counteract the deleterious effects of Aβ_25–35_ on cell viability in both cell lines. On the contrary, 1 μM FD22a induced a significant increase in cell viability compared to Aβ_25–35_-treated cells (Figure 3C,D). For these reasons, the subsequent experiments were carried out using 1 μM FD22a.

### 3.2. FD22a Inhibits β-Amyloid-Mediated Release of Proinflammatory Factors in HMC3 and U87-MG Cells

Microglia represent the first line of immune defense within the CNS, and microglial dysfunction is considered a pathogenic mechanism common to several neurological disorders [38]. Studies have revealed that Aβ peptides activate microglia to release a large variety of proinflammatory factors [39,40]. Therefore, a potential strategy to delay Alzheimer’s disease (AD) onset and possibly prevent its progression could be suppressing the response of microglial cells to inflammatory stress [41,42]. Based on these premises, in our study we used human HMC3 microglial cells to investigate the protective effect of the newly developed CB2R bitopic/dualsteric ligand FD22a against β-amyloid’s cytotoxic and proinflammatory effects. Having assessed that FD22a (1 μM) can protect HMC3 cells from β-amyloid-induced cytotoxicity, we went on to evaluate the ability of FD22a to withstand the increased production of proinflammatory cytokines induced by β-amyloid. In HMC3 cells, Aβ_25–35_ (1 μM for 48 h) promoted a significant increase in the release of common proinflammatory cytokines (TNFα and IL-6) (Figure 4A,B), whereas no effect on the release of anti-inflammatory cytokine IL-10 was observed (Figure 4C). In all experiments, treatment with FD22a (1 μM) alone did not show any significant change as compared to control cells. Then, we repeated the experiments, exposing HMC3 cells to pretreatment with 1 μM FD22a. Compared with the β-amyloid-treated group, pretreatment with FD22a was demonstrated to significantly counteract the β-amyloid-increased secretion of TNFα and IL-6, even though the level of both cytokines was still higher as compared to control cells (Figure 4A,B). In addition, pretreatment with FD22a followed by Aβ_25–35_ treatment induced a significantly enhanced secretion of anti-inflammatory cytokine IL-10 (Figure 4C). Taken together, these findings indicated the potential of FD22a to inhibit β-amyloid-induced microglial activation.

Since the role of CB2R in the beneficial effects of FD22a on the inflammatory response of LPS/TNFα-treated HMC3 cells has been previously reported, we additionally examined the involvement of CB2R in mediating the anti-inflammatory properties of FD22a in β-amyloid-induced HMC3 cells. As shown in Figure 4A–C, co-administration of CB2R selective antagonist SR144528 (1 μM) almost completely abolished the protective effect of FD22a against β-amyloid-induced microglial activation. The same set of experiments was also carried out in U87-MG cells, revealing that the FD22a/CB2R system was able to efficiently suppress β-amyloid-induced enhanced production of proinflammatory cytokines, namely TNFα and IL-6 (Figure 5A,B), and to promote the secretion of anti-inflammatory cytokine IL-10 (Figure 5C). As detected in HMC3 cells, treatment with FD22a (1 μM) alone did not produce any significant effect on cytokine release in U87-MG cells.

### 3.3. FD22a Prevents β-Amyloid-Induced Down-Regulation of Autophagy in U87-MG Cells

ATG is crucial for neuronal homeostasis, and its dysfunction has been directly linked to a growing number of NDDs. Accordingly, the induction of ATG may be exploited as a strategy to assist neurons to survive by clearing abnormal protein aggregates [43]. The kinase mammalian target of rapamycin (mTOR) is a central modulator of ATG [44], and a marked up-regulation of mTOR is known to contribute to AD progression in humans [45]. Hence, U87-MG cells, which feature a similar mTOR up-regulation leading to ATG suppression [46,47,48,49], may represent the ideal cell line for in vitro studies.

Having assessed that FD22a (1 μM) was able to protect U87-MG cells from β-amyloid-induced cytotoxicity and proinflammatory response, we went on to examine whether exposure of U87-MG cells to Aβ_25–35_ could lead to further depression of autophagy and whether pretreatment with FD22a could resolve, or at least attenuate, the deleterious effect of β-amyloid in such cells.

Gene expression analysis revealed that exposure of U87-MG cells to 10 μM Aβ_25–35_ for 48 h leads to a significant decrease in the expression of proautophagy-related genes, such as microtubule-associated protein 1 light chain 3 (LC3), sirtuin 1 (SIRT1), and sirtuin 6 (SIRT6), and to increase expression of negative regulator of ATG such as mTOR, sigma-1 receptor (SIGMAR1), and sirtuin 5 (SIRT5) (Figure 6A). Increased expression of inflammatory-response-related genes, such as monocyte chemoattract protein-1 (MCP1) and transcription factor NF-κB, was also observed (Figure 6B). Pretreatment with FD22a (1 μM) for 24 h was demonstrated to efficiently prevent the depression of autophagy (Figure 6A) and the proinflammatory response caused by β-amyloid exposure (Figure 6B). Notably, treatment with FD22a (1 μM) alone was not able to produce any significant transcriptional effect in U87-MG cells.

### 3.4. Quantitative Proteomic Analysis Uncovers the Activation of TFEB in the Restoration of Autophagy by FD22a

To uncover potential molecular pathways and protein targets involved in FD22a’s effect and in its potential protective action against the deleterious impact caused by β-amyloid exposure, we performed a quantitative proteomic analysis of U87-MG cells. The 2DE protein maps of cellular protein extracts obtained from U87-MG cells in different treatment conditions were compared.

Overall, an average of 1800 ± 70 spots were found within a linear pH range from 3 to 10. A representative gel image is shown in Figure 7A (representative 2DE images of all groups are shown in Appendix A), whereas the Venn diagram shown in Figure 7B shows the number of protein spots found significantly differentially expressed in different comparisons.

Regarding comparison with the control, 67 spots were modified by the treatment with FD22a and 41 by the treatment with Aβ_25–35_, whereas pretreatment with FD22a reduced protein spot changes, as suggested by the FD22a + Aβ_25–35_ vs. Aβ_25–35_ comparison. A volcano plot was constructed to represent fold change and *p*-value in protein expression for this comparison. It revealed that about 50 percent of the 20 spots found with a significant change in expression were up-regulated (Figure 7C).

These spots were identified by LC-MS/MS. The name of the identified proteins, the molecular weight (MW), isoelectric point (pI), score, coverage values of MS/MS, ratio, and p-values are listed in Table 2, whereas Figure 8 shows box plots of data distribution obtained for these proteins in different treatment conditions. Compared with the Aβ_25–35_ group, the combined FD22a and Aβ_25–35_ treatment significantly restores to the control value the expression of HS90A/B, lamin A (LMNA), fructose biphosphate-aldolase (ALDOA), calcyclin-binding protein (CYBP), ubiquitin carboxyl-terminal hydrolase isozyme L1 (UCHL1), and peroxiredoxin 2 (PRDX2). Moreover, Table 3 and Table 4 report the proteins that were found differentially expressed when comparing both FD22a and Aβ_25–35_ treatment with control experiments, respectively.

To explore the molecular pathways involved in FD22a action and its ability to protect against the deleterious effects of Aβ_25–35_, proteins that were found to express differently were analyzed by Ingenuity Pathway Analysis (IPA, QIAGEN Redwood City, CA, USA, www.qiagen.com/ingenuity). An involvement of the CLEAR signaling pathway (z score = 2.2) with a TFEB-dependent activation of autophagy and lysosome biogenesis emerged from the analysis of proteins differently expressed in relation to the treatment with FD22a. Accordingly, an inhibition of NF-κB was observed in the network analysis (Figure 9).

Moreover, a list of potential regulators was obtained from causal network analysis, with significant positive and negative z-scores (Appendix A): among these, the activation of Next to BRCA1 gene 1 protein (NBR1) (z-score = 3.4, *p*-value = 1.16 × 10^−8^), a ubiquitin-binding autophagy adapter which acts as a receptor for selective autophagosomal degradation of ubiquitinated targets, agreed with induction of autophagy by FD22a. On the other hand, IPA of proteins differently expressed after exposure of U87-MG cells to Aβ_25–35_ supported a loss of autophagic response with caspase activation and inhibition of sestrin 2 (SESN2) (z-score = −2.1; *p*-value = 0.00002), a stress-inducible protein, able to activate the specific autophagic machinery for degradation of mitochondria (Appendix A).

Studies have suggested that the dephosphorylation of TFEB at serine 211 (TFEB-S211) promoted the nuclear entry of TFEB, which regulates the expression of autophagy-related genes [2,50].

Since the IPA suggested that the TFEB pathway could be involved in the action of FD22a, we performed Western blot analysis to evaluate the expression of total TFEB and TFEB-S211 in different treatment conditions. Western blot analysis revealed that exposure of U87-MG cells to 10 μM Aβ_25–35_ for 48h led to a significant increase in TFEB-S211 expression (4.2-fold with respect to control, *p*-value < 0.001), and pretreatment with 1 μM FD22a attenuated the expression of the phosphorylated form by about 26% (Figure 10).

Western blot analysis also revealed that Aβ_25–35_ treatment produced an increased expression of the phosphorylated form of mTOR (p-mTOR) in comparison to control cells, whereas the addition of FD22a induced an approximately 33% reduction in mTOR phosphorylation with respect to Aβ_25–35_-treated cells (*p*-value = 0.048) (Figure 10), thus confirming gene expression results. A reduction of p-mTOR, the active form of mTOR (Figure 10), and an increase in phosphatase expression observed in 2DE (Table 2) may concur with the dephosphorylation of TFEB induced by FD22a.

## 4. Discussion

In this study, we investigated the protective effect of the newly developed CB2R bitopic/dualsteric ligand FD22a against the toxicity of β-amyloid (Aβ_25–35_ peptide) on human cellular models of neurodegeneration, including microglial (HMC3) and glioblastoma (U87-MG) cell lines. Our results showed that FD22a protected and rescued both cell lines from β-amyloid cytotoxic and proinflammatory effects and specifically prevented β-amyloid-induced depression of the autophagy–lysosome pathway (ALP) in U87-MG cells by promoting a TFEB-dependent activation of autophagy and lysosome biogenesis. Given the strong evidence for ALP’s impairment in all major forms of NDDs [51,52,53], drugs capable of restoring autophagy deficits hold significant potential in the treatment of NDDs.

Our gene expression data revealed that exposure of U87-MG cells to β-amyloid leads to significantly decreased expression of proautophagy marker genes (i.e., LC3, SIRT1, and SIRT6) and increased expression of negative autophagy regulators, such as mTOR, SIGMAR1, and SIRT5. Increased expression of inflammatory-response-related genes, including MCP1 and NF-κB, was also observed. Notably, pretreatment with FD22a prevented the depression of autophagy and the proinflammatory response caused by β-amyloid exposure, strengthening previous evidence of neuroprotective activity described for FD22a [27] and supporting a beneficial role of CB2R activation in β-amyloid-dependent neuroinflammation, as seen in AD pathology [23]. To explore in more detail the molecular pathways involved in the protective action of FD22a against the deleterious effects induced by β-amyloid, we performed a quantitative proteomic analysis of U87-MG cells. Proteins that were found to express differently following different treatment conditions, such as control, exposure to β-amyloid alone, and co-treatment with FD22a and β-amyloid, were analyzed by IPA to identify specific networks. Molecular pathway analysis revealed that in U87-MG cells’ FD22a treatment significantly impacts the CLEAR signaling pathway, promoting a TFEB-dependent activation of autophagy and lysosome biogenesis. Accordingly, inhibition of NF-κB and activation of Next to BRCA1 gene 1 protein (NBR1), with the latter acting as a receptor for selective autophagosomal degradation of ubiquitinated targets, were observed, supporting the induction of autophagy by FD22a. On the other hand, after exposure of U87-MG cells to β-amyloid, network analysis supported a loss of autophagic response with caspase activation and inhibition of the stress-inducible metabolic protein sestrin 2 (SESN2), which is widely recognized as a key regulator of cellular homeostasis [52,54]. Furthermore, compared with the Aβ_25–35_ treatment group, the combined FD22a and Aβ_25–35_ treatment significantly restores, to the control value, the expression of HS90A/B, lamin A (LMNA), fructose bisphosphate-aldolase (ALDOA), calcyclin-binding protein (CYBP), ubiquitin carboxyl-terminal hydrolase isozyme L1 (UCHL1), and peroxiredoxin 2 (PRDX2), collectively contributing to the recovery of cell viability and vitality and suggesting a possible role of CB2R activation in promoting cell survival in the face of β-amyloid toxic insults.

In particular, the significant reduction in cofilin-1 expression actually could derive from CBR2 activation. Cofilin is an essential actin regulatory protein that constitutively severs actin filaments, and its activation is often an early event in cell migration. CBR2 agonists have been shown to modulate the activities of cofilin [55], and Yang et al. suggested that anti-inflammatory effects of CBR2 agonists may be mediated by cofilin-1 protein [56].

Numerous studies have shown that improving intracellular clearance may alleviate the symptoms associated with a large variety of NDDs. Therefore, regulating TFEB activity may be a promising therapeutic strategy against NDDs. The main mechanisms involved in the regulation of TFEB are predominantly posttranslational modifications, including phosphorylation, acetylation, SUMOylating, PARsylation, and glycosylation [50]. Among them, the phosphorylation of diverse serine residues plays a central role because it maintains TFEB in the cytoplasm, preventing its nuclear entry and activation. The principal negative regulator of TFEB activity is the mammalian target of rapamycin (mTorc1), which phosphorylates TFEB at at least three serines, S122, S142, and S211 [57,58]. We demonstrated here that exposure of U87-MG cells to Aβ_25–35_ could lead to a significant increase in TFEB-S211 expression, while pretreatment with FD22a significantly attenuated the expression of the phosphorylated form. In accordance with gene expression results, Western blot analysis also revealed that Aβ_25–35_ treatment produced an increased expression of the phosphorylated form of mTOR (p-mTOR) in comparison to control cells, whereas the addition of FD22a induced a significant reduction of p-mTOR with respect to control and Aβ_25–35_-treated cells. Notably, the detected reduction of p-mTOR and increased expression of phosphatases revealed in 2DE experiments may concur with the dephosphorylation of TFEB induced by FD22a.

## 5. Conclusions

Overall, the results of our study highlight the potential for a multitarget treatment of neurodegenerative pathologies via FD22a-mediated activation of CB2R. The observed ability of FD22a to promote TFEB nuclear entry by dephosphorylation of S211, the target serine of mTOR, followed by TFEB-mediated activation of autophagic lysosomal function, associated with the ability to prevent β-amyloid-induced cytotoxic and proinflammatory effects, may have a marked relevance in the prevention and/or treatment of AD pathology. Even though our work provides convincing evidence of the potential of FD22a to target neurodegeneration, extensively illustrating the ability of this novel CB2R bitopic ligand to efficiently counteract the deleterious effects of β-amyloid in human glial cells, further investigations on in vivo models of AD and knockout of CB2R will be necessary to corroborate the therapeutic potential of CB2R activation in slowing or reversing AD. Furthermore, a detailed investigation of FD22a’s pharmacokinetic properties will also be fundamental to pursue a future therapeutic application of this novel CB2R bitopic ligand.

## Figures and Tables

**Figure 1 cells-13-00875-f001:**
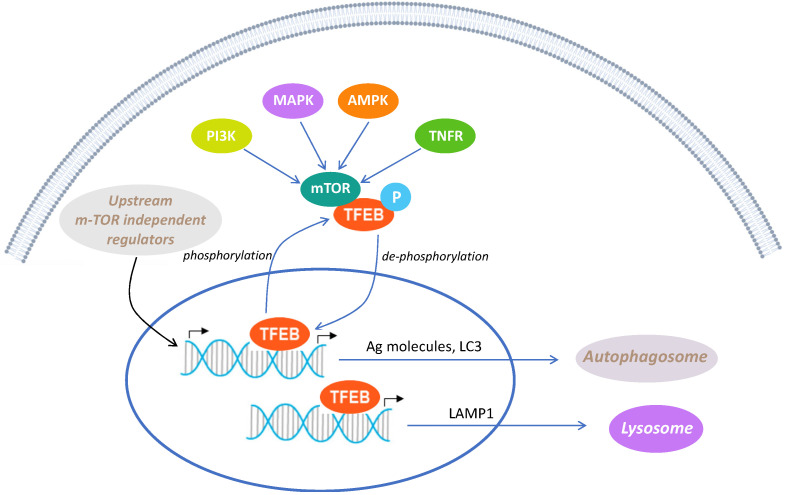
Regulation of TFEB activity, a key transcription molecule regulating autophagy. Intracellular molecules, including PI3K, MAPK, AMPK, and TNFR, possess the capability to activate the mTOR pathway, a prominent negative regulator of autophagy. mTOR, functioning as a kinase, plays a crucial role in regulating the localization and activation of TFEB, a crucial transcription factor, which governs autophagy at the transcriptional level by promoting the expression of multiple lysosomal genes, influencing autolysosome production and function. Specifically, activated mTOR phosphorylates TFEB, inhibiting its activity and confining it to the cytoplasm. Inhibiting mTOR leads to TFEB dephosphorylation, enabling its nuclear translocation and transcriptional activity. Within the nucleus, TFEB regulates autophagy in two ways: promoting the expression of autophagy-related molecules for enhanced autophagy, and activating lysosomal-pathway-related molecules, especially LAMP1, to foster lysosome formation. This orchestrated process facilitates the degradation of damaged organelles, recycling their products, such as amino acids, for cellular benefit.

**Figure 2 cells-13-00875-f002:**
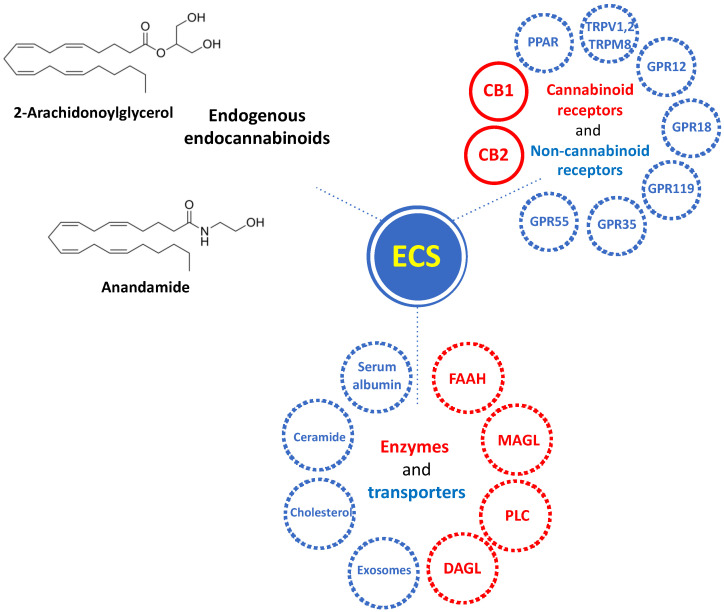
Overview of the components of the endocannabinoid system (ECS). ECS consists in endogenous endocannabinoids Anandamide (AEA) and 2-Arachidonoylglycerol (2-AG); cannabinoid receptors type 1 (CB1) and type 2 (CB2), and non-cannabinoid receptors GPR55, GPR35, GPR119, GPR18, GPR12, TRPM8, TRPV1, TRPV2, PPAR; enzymes for endocannabinoid synthesis (diacylglycerol lipase (DAGL) and phospholipase C (PLC)) and degradation (monoacylglycerol lipase (MAGL) and fatty acid amide hydrolase (FAAH)), and transporters, including serum albumin, ceramide, cholesterol.

**Figure 3 cells-13-00875-f003:**
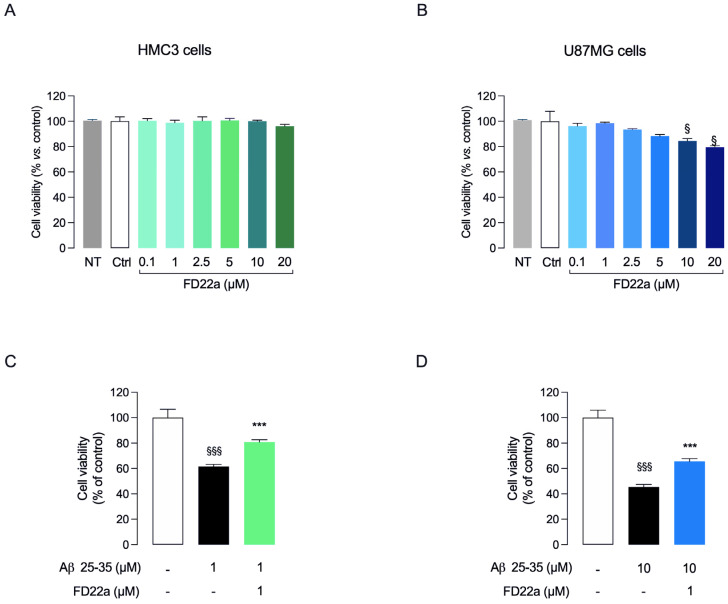
FD22a prevents Aβ_25–35_-induced damage in HMC3 and U87-MG cell lines. (**A**) HMC3 and (**B**) U87-MG cells were treated with FD22a (0.1–20 μM) for 24 h; (**C**) HMC3 and (**D**) U87-MG cells were pretreated with FD22a (1 μM) and after 24 h exposed respectively to 1 μM or 10 μM Aβ_25–35_ for 48 h. In both experimental models, cell viability was quantified by MTT assay. Each bar corresponds to the means ± SEM of at least four independent experiments. Data were analyzed by one-way analysis of variance (ANOVA) followed by Dunnett’s test. ^§^
*p* < 0.05 with respect to vehicle-treated cells (Ctrl). ^$$$^ *p* < 0.005 respect to Ctrl; *** *p* < 0.005 with respect to Aβ_25–35_-treated cells.

**Figure 4 cells-13-00875-f004:**
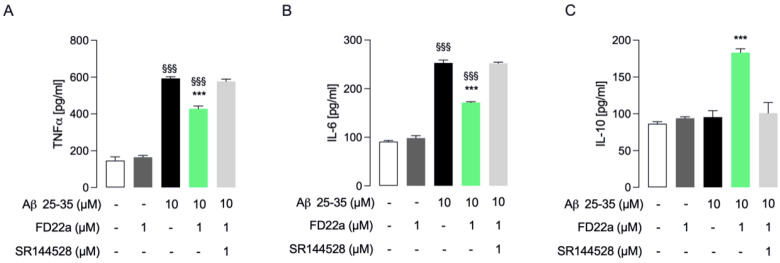
FD22a/CB2R system modulates Aβ_25–35_-mediated secretion of inflammatory cytokines in HMC3 cells. In HMC3 cells, FD22a pretreatment counteracts the release (pg/mL) of proinflammatory TNFα (**A**) and IL-6 (**B**) induced by Aβ_25–35_ and stimulates the secretion of anti-inflammatory IL-10 (**C**). Each bar corresponds to the means ± SEM of at least three independent experiments. Data were analyzed by one-way analysis of variance (ANOVA) followed by Dunnett’s test. ^$$$^ *p* < 0.005 with respect to vehicle-treated cells (control cells); *** *p* < 0.005 with respect to Aβ_25–35_-treated cells.

**Figure 5 cells-13-00875-f005:**
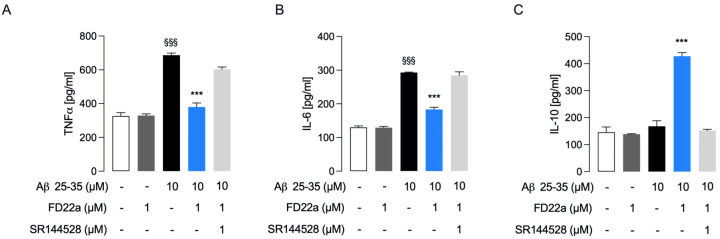
FD22a/CB2R system modulates Aβ_25–35_-mediated release of inflammatory cytokines in U87-MG cells. In U87-MG cells, FD22a pretreatment counteracts the release (pg/mL) of proinflammatory TNFα (**A**) and IL-6 (**B**) induced by Aβ_25–35_ and stimulates the secretion of anti-inflammatory IL-10 (**C**). Each bar corresponds to the means ± SEM of at least three independent experiments. Data were analyzed by one-way analysis of variance (ANOVA) followed by Dunnett’s test. ^$$$^ *p* < 0.005 with respect to vehicle-treated cells (control cells); *** *p* < 0.005 with respect to Aβ_25–35_-treated cells.

**Figure 6 cells-13-00875-f006:**
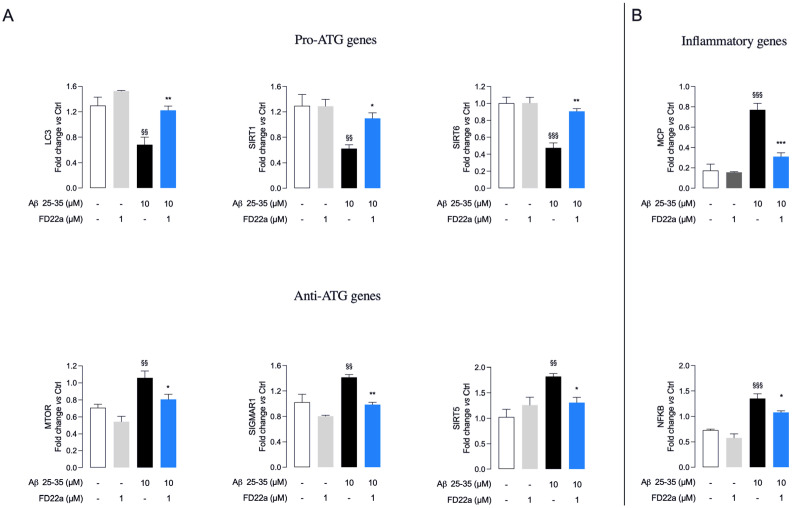
FD22a pretreatment prevents Aβ_25–35_ deleterious transcriptional effects on autophagy and inflammation. Pretreatment with FD22a significantly modulates transcriptional expression of selected autophagy (ATG)-related (**A**) and proinflammatory (**B**) genes. Each bar corresponds to the means ± SEM of at least three independent experiments. Data were analyzed by one-way analysis of variance (ANOVA) followed by Tukey’s test. ^$$^ *p* < 0.01 with respect to vehicle-treated cells (control cells) ^$$$^ *p* < 0.005 with respect to vehicle-treated cells (control cells); * *p* < 0.05 with respect to Aβ_25–35_-treated cells; ** *p* < 0.01 with respect to Aβ_25–35_-treated cells; *** *p* < 0.005 with respect to Aβ_25–35_-treated cells.

**Figure 7 cells-13-00875-f007:**
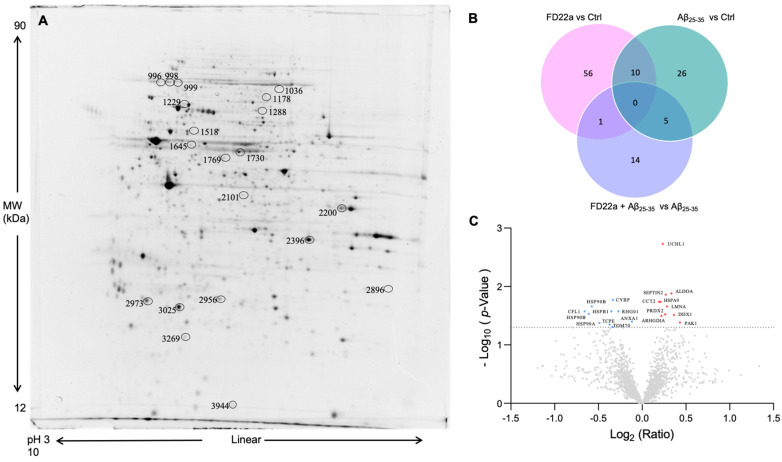
(**A**) Representative 2DE image of U87-MG proteome. Protein extracts were separated in a linear pH 3–10 gradient. SDS-PAGE was performed using 12% acrylamide. Gels were stained with fluorescent dye and acquired by ImageQuant TL 7. (**B**) Venn diagram showing the number of proteins found differentially expressed in the different comparisons: FD22a vs. Ctrl, Aβ_25–35_ vs. Ctrl, and FD22a + Aβ_25–35_ vs. Aβ_25–35_. Both unique and overlapping proteins are reported as numbers (Venny 2.0.2). (**C**) Scatter plot of fold change (*x*-axis) against log_10_
*p*-value (*y*-axis) of quantified proteins obtained for FD22a + Aβ_25–35_ vs. Aβ_25–35_ comparison. Up-regulated and down-regulated proteins are colored red and blue, respectively. Only proteins that showed both *p*-value and q-value < 0.05 were identified. Dotted line indicates the threshold of significance. The gene names of identified proteins are shown in the scatter plot.

**Figure 8 cells-13-00875-f008:**
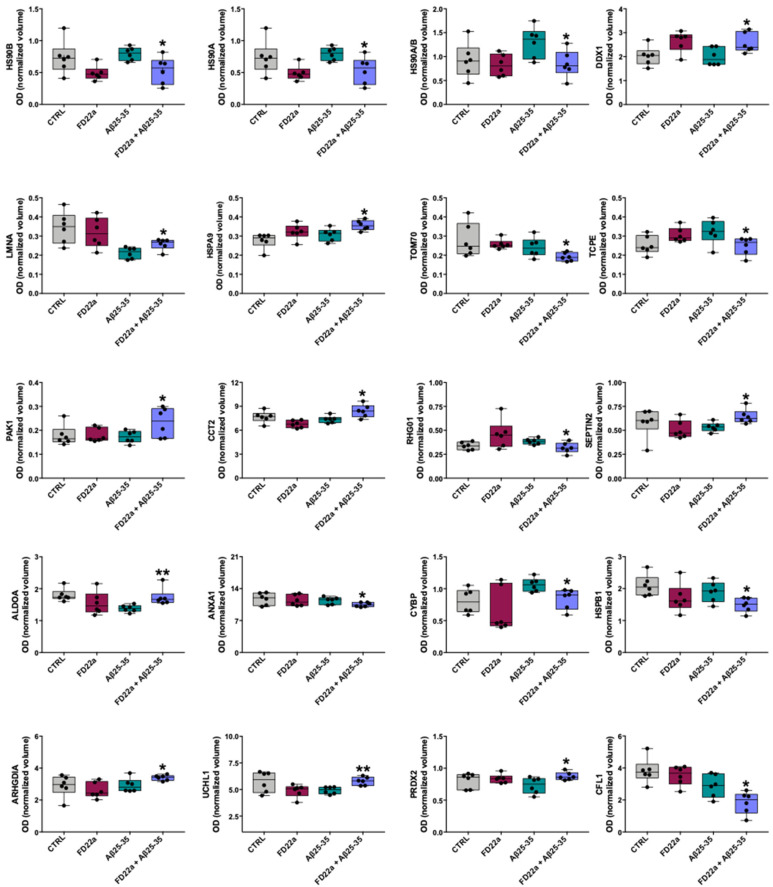
Box plots of proteins found differentially expressed in the comparison FD22a + Aβ_25–35_ vs. Aβ_25–35_. The optical density (OD) of normalized spot volume of the six biological replicates is shown for different conditions (control, FD22a, Aβ_25–35_, FD22a + Aβ_25–35_). Data were analyzed by Mann–Whitney test. * *p* < 0.05, ** *p* < 0.01, with respect to Aβ_25–35_.

**Figure 9 cells-13-00875-f009:**
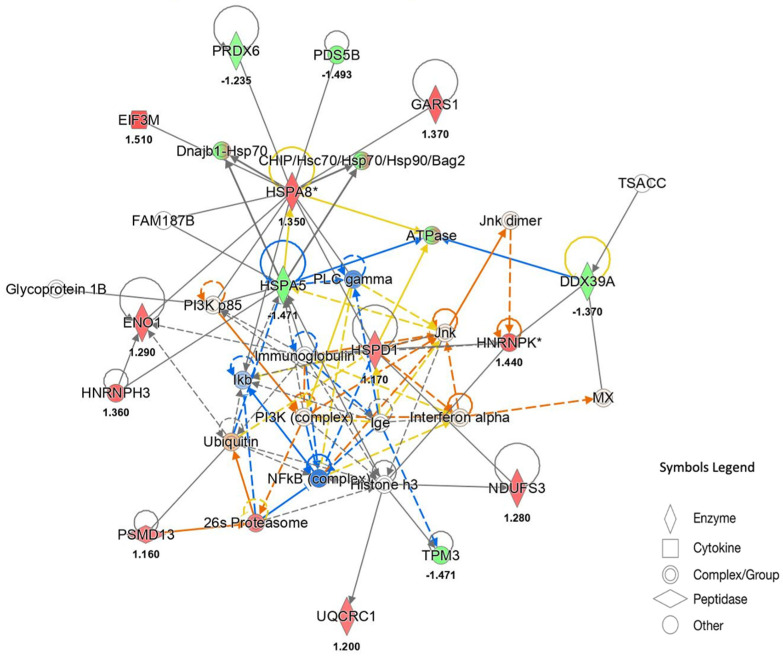
Functional network derived from QIAGEN’s Ingenuity Pathway Analysis of proteins differentially expressed in the FD22a + Aβ_25–35_ vs. Aβ_25–35_ comparison. The network describes functional relationships among proteins based on known associations in the literature. Solid line: direct interaction; dotted line: indirect interaction. Red and green indicate up- and down-regulated proteins, respectively. Orange suggests an activation whereas blue suggests an inhibition. (*) This protein has been identified in many spots. The number below the protein symbol indicates the fold change value of expression.

**Figure 10 cells-13-00875-f010:**
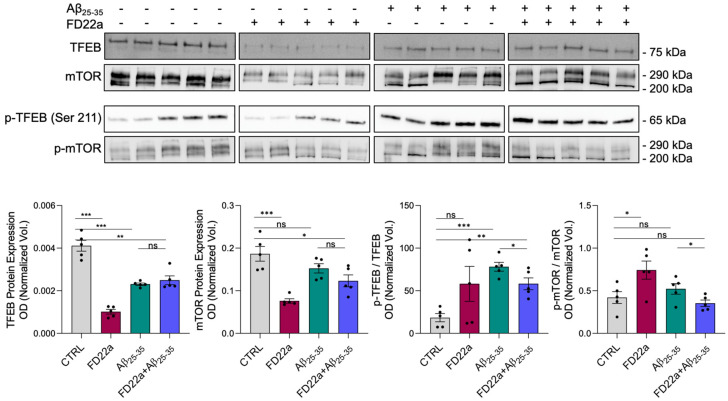
Western blot detection and quantification of TFEB, p-TFEB (S211), mTOR, and p-mTOR in U87-MG control cells, after the addition of FD22a and after treatment with Aβ_25–35_ alone and in the presence of FD22a. Each bar graph represents the mean ± SEM of five independent experiments. Optical density of each immunoreactive band was normalized on total protein obtained from RuBP staining. For p-TFEB and p-mTOR the expression level of TFEB and mTOR, respectively, were used as loading control. An unpaired *t*-test was used to compare differences among treatments (Prism 7; GraphPad Software, San Diego, CA, USA), * *p* < 0.05, ** *p* < 0.01, and *** *p* < 0.001. ns means not significant.

**Table 1 cells-13-00875-t001:** Primer sequences for Real-Time PCR experiments.

Reference Sequence (RefSeq) RNA	Gene Symbol	Primer Sequences
NM_002046	GAPDH	(F) 5′-CCCTTCATTGACCTCAACTACATG
		(R) 5′-TGGGATTTCCATTGATGACAAGC
NM_022818.5	LC3	(F) 5′-AGTCTTCTCTTCAGGTTCAC
		(R) 5′-CTCACACAGCCCGTTTAC
NM_004958.4	MTOR	(F) 5′-TGCCTTCACAGATACCCAG
		(R) 5′-AGACCTCACAGCCACAGA
NM_001282208	SIGMAR1	(F) 5′-CTTCTACCCAGGGGAGAC
		(R) 5′-GCATAGGAGCGAAGAGTAT
NM_001314049.2	SIRT1	(F) 5′-GGGTTCTTCTAAACTTGGACTCT
		(R) 5′-GTAGGCGGCTTGATGGTAAT
NM_054354940.1	SIRT5	(F) 5′-CAAATCTGGTTTCGTGTGGAC
		(R) 5′-AATAACTAAAGCCCGCCTCAA
NM_001193285	SIRT6	(F) 5′-CTCCTCCGCTTCCTGGTC
		(R) 5′-TTACACTTGGCACATTCTTCC
NM_002982.4	MCP1	(F) 5′-GAGAGGCTGAGACTAACC
		(R) 5′-TGATTGCATCTGGCTGAG
NM_001404662	NFKB	(F) 5′-CCTTTCTCATCCCATCTTT
		(R) 5′-CCTCAATGTCCTCTTTCTG

**Table 2 cells-13-00875-t002:** List of proteins found differentially expressed in the comparison of U87-MG cells treated with FD22a + Aβ_25–35_ vs. cells treated with Aβ_25–35_ identified by LC-MS/MS. ID: SwissProt accession number, MW: molecular weight, pI: isoelectric point.

#	ID	Gene	Protein Name	Score	Cov	Pep	pI	MW	*p*-Value	Ratio FD22a + Aβ_25–35_/Aβ_25–35_
996	P08238	HS90B	Heat shock protein HSP 90-beta	70	9	6	4.97	83,543	0.005	0.67
998	P07900	HS90A	Heat shock protein HSP 90-alpha	72 *	3	1	4.94	84,660	0.022	0.71
999	P07900	HS90A	Heat shock protein HSP 90-alpha	73 *	3	1	4.94	84,660	0.036	0.65
999	P08238	HS90B	Heat shock protein HSP 90-beta	73 *	3	1	4.96	83,264	0.036	0.65
1036	Q92499	DDX1	ATP-dependent RNA helicase	84 *	5	4	6.8	82,432	0.076	1.29
1178	P02545	LMNA	Prelamin-A/C	159 *	18	12	6.57	74,140	0.005	1.22
1229	P38646	HSPA9	Stress-70 protein. mitochondrial	73	11	6	5.87	73,920	0.008	1.16
1288	O94826	TOM70	Mitochondrial import receptor subunit TOM70	45 *	5	2	6.82	67,455	0.038	0.79
1518	P48643	TCPE	T-complex protein 1 subunit epsilon	40 *	2	1	6.1	60,534	0.143	0.77
1645	Q13153	PAK1	Serine/threonine-protein kinase PAK 1	59	19	7	5.55	60,894	0.023	1.35
1730	P78371	CCT2	T-complex protein 1 subunit beta	150	32	11	6.01	57,794	0.000	1.14
1769	Q07960	RHG01	Rho GTPase-activating protein 1	36 *	3	1	5.85	50,436	0.249	0.83
2101	Q15019	SEPTIN2	Septin-2	64	18	4	6.15	41,689	0.023	1.21
2200	P04075	ALDOA	Fructose-bisphosphate aldolase A	139 *	9	9	8.39	39,420	0.001	1.26
2396	P04083	ANXA1	Annexin A1	174	39	11	6.57	38,918	0.004	0.92
2896	Q9HB71	CYBP	Calcyclin-binding protein	59 *	7	2	8.32	26,210	0.005	0.79
2956	P04792	HSPB1	Heat shock protein beta-1	66	16	4	5.98	22,826	0.006	0.78
2973	P52565	ARHGDIA	Rho GDP-dissociation inhibitor 1	78	29	6	5.02	23,250	0.009	1.16
3025	P09936	UCHL1	Ubiquitin carboxyl-terminal hydrolase isozyme L1	56	19	3	5.33	25,151	0.001	1.18
3269	P32119	PRDX2	Peroxiredoxin-2	70	22	4	5.66	22,049	0.084	1.20
3944	P23528	CFL1	Cofilin-1	120	48	7	8.22	18,719	0.001	0.63

*—10lg*p*-value of samples analyzed by Orbitrap.

**Table 3 cells-13-00875-t003:** List of proteins found differentially expressed in the comparison of U87-MG cells treated with FD22a vs. cells without treatment identified by LC-MS/MS. ID: SwissProt accession number, MW: molecular weight, pI: isoelectric point.

#	ID	Gene	Protein Name	Score	Cov	Pep	pI	MW	*p*-Value	Ratio FD22a/Ctrl
858	P12814	ACTN1	Alpha-actinin-1	82	11	8	5.25	103,563	0.0107	0.70
914	P55072	VCP	Transitional endoplasmic reticulum ATPase	190	28	18	5.14	89,950	0.0478	0.69
1140	Q9NTI5	PDS5B	Sister chromatid cohesion protein PDS5 homolog B	56	7	9	8.67	16,5818	0.0129	0.67
1161	P11021	HSPA5	Endoplasmic reticulum chaperone BiP	237	36	18	5.07	72,402	0.0035	0.68
1199	P41250	GARS1	Glycine--tRNA ligase	116	21	10	6.61	83,854	0.0010	1.37
1205	P26038	MSN	Moesin	87	14	8	6.08	67,892	0.0242	1.48
1278	P11142	HSPA8	Heat shock cognate 71 kDa protein	89	14	7	5.37	71,082	0.0273	1.28
1301	P11142	HSPA8	Heat shock cognate 71 kDa protein	55	9	5	5.37	71,082	0.0184	1.35
1317	O75864	PPP1R37	Protein phosphatase 1 regulatory subunit 37	47	6	3	4.97	74,767	0.0435	0.83
1421	P61978	HNRNPK	Heterogeneous nuclear ribonucleoprotein K	78	20	7	5.39	51,230	0.0017	1.44
1432	P61978	HNRNPK	Heterogeneous nuclear ribonucleoprotein K	150	25	10	5.39	51,230	0.0457	1.26
1435	Q9H6N6	MYH16	Putative uncharacterized protein MYH16	64	14	12	5.4	128,439	0.0051	0.76
1443	P61978	HNRNPK	Heterogeneous nuclear ribonucleoprotein K	96	23	8	5.39	51,230	0.0069	1.33
1451	Q16555	DPYSL2	Dihydropyrimidinase-related protein 2	58	14	5	5.95	62,711	0.0014	0.77
1461	P48643	TCPG	T-complex protein 1 subunit gamma	93 *	4	2	6.1	60,534	0.0076	0.82
1461	Q16555	DPYL2	Dihydropyrimidinase-related protein 2	54 *	2	1	5.95	62,294	0.0076	0.82
1511	P08670	VIM	Vimentin	147	39	14	5.06	53,676	0.0274	0.78
1524	P10809	HSPD1	60 kDa heat shock protein. mitochondrial	203	41	16	5.7	61,187	0.0308	1.17
1610	P08670	VIM	Vimentin	221	48	21	5.06	53,676	0.0156	1.20
1641	P68371	TUBB4B	Tubulin beta-4B chain	142	31	13	4.79	50,255	0.0043	0.51
1725	O43175	PHGDH	D-3-phosphoglycerate dehydrogenase	56	12	6	6.29	57,356	0.0005	1.81
1726	P28329	CHAT	Choline O-acetyltransferase	73	9	8	8.9	83,852	0.0282	0.74
1730	P78371	CCT2	T-complex protein 1 subunit beta	150	32	11	6.01	57,794	0.0271	0.88
1745	O00148	DDX39A	ATP-dependent RNA helicase	92	19	8	5.46	49,611	0.0263	0.73
1848	P48594	SERPINB4	Serpin B4	60	12	5	5.86	44,997	0.0192	1.20
1859	P06733	ENO1	Alpha-enolase	61	17	5	7.01	47,481	0.0479	1.29
1864	P08670	VIM	Vimentin	70	29	12	5.06	53,676	0.0056	0.48
1877	P31930	UQCRC1	Cytochrome b-c1 complex subunit 1. mitochondrial	84	27	7	5.94	53,297	0.0397	1.20
1951	P08670	VIM	Vimentin	123	24	10	5.06	53,676	0.0117	0.40
2042	O75874	IDHC	Isocitrate dehydrogenase [NADP] cytoplasmic (IDH)	124 *	17	7	6.53	45,659	0.0491	0.85
2077	Q7L2H7	EIF3M	Eukaryotic translation initiation factor 3 subunit M	62	15	4	5.41	42,932	0.0091	1.51
2140	Q9UNM6	PSMD13	26S proteasome non-ATPase regulatory subunit 13	86	18	5	5.53	43,203	0.0492	1.16
2365	P67775	PPP2CA	Serine/threonine-protein phosphatase 2A catalytic subunit alpha isoform	66	17	4	5.3	36,142	0.0425	1.24
2464	P31942	HNRNPH3	Heterogeneous nuclear ribonucleoprotein H3	70	19	5	6.37	36,960	0.0228	1.36
2491	P62879	GNB2	Guanine nucleotide-binding protein G(I)/G(S)/G(T) subunit beta-2	71	18	6	5.6	38,048	0.0029	1.43
2595	P09525	ANXA4	Annexin A4	59	15	4	5.84	36,088	0.0169	1.32
2637	P07951	TPM2	Tropomyosin beta chain	62	26	7	4.66	32,945	0.0171	1.49
2653	P06753	TPM3	Tropomyosin alpha 3 chain	70	13	6	4.68	32,987	0.0001	0.68
2720	P62258	YWHAE	14-3-3 protein epsilon	79	31	6	4.63	29,326	0.0014	0.54
2806	P61981	YWHAG	14-3-3 protein gamma	68	27	5	4.8	28,456	0.0000	0.55
2864	P63104	YWHAZ	14-3-3 protein zeta/delta	87	31	6	4.73	27,899	0.0001	0.39
2875	P17480	UBTF	Nucleolar transcription factor 1	59	10	8	5.63	89,692	0.0221	1.34
2884	P31946	YWHAB	14-3-3 protein beta/alpha	88	27	6	4.76	28,179	0.0009	0.65
2895	P04083	ANXA1	Annexin A1	65	16	4	6.57	38,918	0.0012	1.75
2932	O75489	NDUFS3	NADH dehydrogenase [ubiquinone] iron-sulfur protein 3. mitochondrial	74	24	5	6.99	30,337	0.0116	1.28
2997	P30041	PRDX6	Peroxiredoxin-6	94	26	5	6	25,133	0.0046	0.81
4526	P61978	HNRNPK	Heterogeneous nuclear ribonucleoprotein K	95	23	8	5.39	51,230	0.0314	0.76

*—10lg*p*-value of samples analyzed by Orbitrap.

**Table 4 cells-13-00875-t004:** List of proteins found differentially expressed in the comparison of U87-MG cells treated with Aβ_25–35_ vs. cells without treatment identified by LC-MS/MS. ID: SwissProt accession number, MW: molecular weight, pI: isoelectric point.

#	ID	Gene	Protein Name	Score	Cov	Pep	pI	MW	*p*-Value	Ratio Aβ_25–35_/Ctrl
669	P53396	ACLY	ATP-citrate synthase	72 *	2	2	6.95	120,839	0.0151	0.76
669	P53992	SC24C	Protein transport protein Sec24C	63 *	2	2	6.71	118,325	0.0151	0.76
1099	P13798	APEH	Acylamino-acid-releasing enzyme	58	12	7	5.29	82,142	0.0485	1.29
1136	P0CG48	UBC	Polyubiquitin-C	49 *	2	1	7.16	77,039	0.0007	0.43
1140	Q9NTI5	PDS5B	Sister chromatid cohesion protein PDS5 homolog B	56	7	9	8.67	165,818	0.0036	0.43
1149	P02545	LMNA	Prelamin-A/C [Cleaved into: Lamin-A/C (70 kDa lamin) (Renal carcinoma antigen NY-REN-32)]	41 *	2	1	6.57	74,140	0.0185	0.55
1178	P02545	LMNA	Prelamin-A/C	159 *	18	12	6.57	74,140	0.0051	0.62
1179	P41250	GARS	Glycine--tRNA ligase	79 *	3	3	6.61	83,166	0.0144	1.39
1314	P20700	LMNB1	Lamin-B1	80	11	8	5.11	66,653	0.0330	0.74
1317	O75864	PPP1R37	Protein phosphatase 1 regulatory subunit 37	47	6	3	4.97	74,767	0.0078	0.58
1331	Q9NSD9	SYFB	Phenylalanine--tRNA ligase beta subunit	90 *	8	8	6.39	66,116	0.0524	0.65
1356	P31040	SDHA	Succinate dehydrogenase [ubiquinone] flavoprotein subunit mitochondrial	94 *	4	4	7.06	72,692	0.0021	0.54
1432	P61978	HNRNPK	Heterogeneous nuclear ribonucleoprotein K	150	25	10	5.39	51,230	0.1540	1.36
1451	Q16555	DPYSL2	Dihydropyrimidinase-related protein 2	58	14	5	5.95	62,711	0.0002	0.86
1461	P48643	TCPG	T-complex protein 1 subunit gamma	93 *	4	2	6.1	60,534	0.0002	0.87
1461	Q16555	DPYL2	Dihydropyrimidinase-related protein 2	54 *	2	1	5.95	62,294	0.0002	0.87
2042	O75874	IDHC	Isocitrate dehydrogenase [NADP] cytoplasmic	124 *	17	7	6.53	45,659	0.0028	0.86
2200	P04075	ALDOA	Fructose-bisphosphate aldolase A	139 *	9	9	8.39	39,420	0.0015	0.77
2896	Q9HB71	CYBP	Calcyclin-binding protein	59 *	7	2	8.32	26,210	0.0055	1.32
2997	P30041	PRDX6	Peroxiredoxin-6	94	26	5	6	25,133	0.0002	0.81
3944	P23528	CFL1	Cofilin-1	120	48	7	8.22	18,719	0.0009	0.75
4526	P61978	HNRNPK	Heterogeneous nuclear ribonucleoprotein K	95	23	8	5.39	51,230	0.0008	0.78

*—10lg*p*-value of samples analyzed by Orbitrap.

## Data Availability

The data presented in this study are available on request from the corresponding author.

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
