# Peer review of "A Proteomic Approach Identified TFEB as a Key Player in the Protective Action of Novel CB2R Bitopic Ligand FD22a against the Deleterious Effects Induced by β-Amyloid in Glial Cells"

_cells, 2024, doi:10.3390/cells13100875_

Round 1
Reviewer 1 Report
Comments and Suggestions for Authors
The study disclosed that a novel CB2R bitopic ligand, FD22a, effectively prevented β-amyloid-induced cytotoxicity and inflammation in HMC3 and U87MG cells. Moreover, the study demonstrated that FD22a was able to counteract the inhibitory effect of Aβ25-35 on the autophagy-lysosomal pathway in U87MG cells by activating the transcription factor TFEB.
However, there are several points that need correction:
1. To investigate the neuroprotective effects of FD22a, further research in neuronal cells and in vivo models is necessary.
2. Please change the color of the MAPK bubble in Figure 1, and use the full words “phosphorylation” and “dephosphorylation” to avoid ambiguity. What is meant by “Upstream regulator”?
3. Figure 2 is not closely related to the study content and is too simplistic; consider replacing it with a textual description.
4. It is essential to include control groups with only solvent DMSO and only FD22a in all experiments to accurately assess the effects of FD22a.
5. The order of Figure 3C, D and A, B should be adjusted to better present the experimental logic.
6. Section 3.2 can be renamed “FD22a inhibits β-amyloid-mediated release of proinflammatory factors in both HMC3 and U87-MG cells,” and the protective effects should be further validated by stimulating neuronal cells with the cell culture supernatant and measuring cell viability.
7. Figure 6 should classify and display the autophagy-promoting genes and autophagy-negative regulators separately.
8. Figure 7A should show the 2DE images of all groups, and the significantly changed spots in the FD22a + Aβ25-35 vs Aβ25-35 comparison should be marked. Figures 7 and 8 should be combined.
9. There is repetition in the protein names in the second and third rows, as well as the first and fourth rows in Table 2; please also check other relevant data to ensure the accuracy of the experimental results.
10. Please confirm that the results in Figure 10 are derived from the analysis of differentially expressed proteins between the FD22a + Aβ25-35 and Aβ25-35 groups.
11. Figure 11 should also compare the FD22a + Aβ25-35 group with the Aβ25-35 group. There seems to be no difference in the levels of p-TFEB and p-mTOR proteins between these two groups in the western blot image. The authors need to utilize cell immunofluorescence and cytoplasmic/nuclear protein extraction combined with western blot to detect the nuclear localization of TFEB in each group to illustrate that FD22a exerts its protective effects by activating TFEB.
Reviewer 2 Report
Comments and Suggestions for Authors
The article “A Proteomic Approach Identified TFEB as a Key Player in the Neuroprotective Action of Novel CB2R Bitopic Ligand FD22a” by Polini et al has attempted to understand the mode of function of drug FD22a which has been shown to have neuroprotective role and so on neurodegenerative disorders. They tested the protective effect (cytotoxic and pro-inflammatory) of the drug. The paper is well written with sufficient background. The material and method sections are detailed and easy to follow. The following comments need to be included in the manuscript to improve the quality.
1. Line 52-54: having a little more information about how TFEB localization, activities, and function are altered in NDD will be helpful for readers without going back to reference.
2. Figure 1: The factor (MAPK) between PI3K and AMPK on the membrane is not visible in the cartoon. Please fix this. Please change "autofagosome to "Autophagosome". Figure 1 is not very clear in demonstrating the mechanism of overall regulation of TFEB upon mTOR activation subsequent phosphorylation/dephosphorylation. It is advised to modify the image to make it more explanatory.
3. Figure 2 needs to be modified to indicate the three different branches,
4. Section 2.2.3. Please mention the antibody dilution used in the WB.
5. Figure 3 C: The plot corresponding to 0.1 is missing. Also, label the unit of concentration being used.
6. Figure 4 A and B: I disagree with the author's claim that FD22a treatment significantly brings the pro-inflammatory cytokines TNF and Il-6 to normal levels. I can still see at least 1.5-fold above the normal level. Having statistical analysis against each other will be helpful.
7. Figure 6A: Please arrange the figure systemically. Put all pro-ATG and negative regulators together. Such as showing SIRT6 on the right with SIGMR1 makes it difficult to correlate.
8. Lines 469-462 are duplicated in the discussion section (lines 526-529). It is advisable to modify the paragraph.
Reviewer 3 Report
Comments and Suggestions for Authors
The manuscript extends the work done by the group published in Ferrisi et al. 2023 and Gado et al. 2022 about the dualsteric ligand FD22a. It covers the thorough proteomic analysis of its action against aBeta 25-35 pathology in 2 cell lines.
I have some questions and points:
line138: The sequnce of Aβ25–35 should be given
It is not clear, whether the Aβ oligomers were characterized or quality controlled by some biophysical method (DLS, AFM,...).
It is not discussed, why these two cell lines were used and whether these are a good model of NDD.
Reviewer 4 Report
Comments and Suggestions for Authors
To explore the potential of FD22a, the first CB2R bitopic/dualsteric ligand, as multitarget neuroprotective drug, the authors investigated its ability to prevent the toxic effect of β-amyloid (Aβ25-35 peptide) on human cellular models of neurodegeneration, HMC3 and U87MG cell lines. They demonstrated that FD22a could efficiently prevent Aβ25-35 cytotoxic and pro-inflammatory effects and stimulate the autophagy–lysosomal pathway by activating its master transcriptional regulator TFEB. However, there are some major issues need to be addressed.
Comments:
1. The authors need to share all the raw files.
2. To validate that TFEB Ser211 phosphorylation is the key for the restoration of autophagy by FD22a, it would be better to perform gene depletion of TFEB followed by rescue experiment with wild-type and Ser211 mutant TGEB, and then test its effect on FD22a stimulated autophagy.
3. To optimize of the FD22a concentration, more concentration should be tested between 1 uM and 10 uM, such as 2 uM, 4 uM, 6 uM and 8 uM.
4. For Figure 3C, the part for o.1uM FD22a treatment was missing.
5. For mass data analysis, how to determine whether the proteins were up- or down-regulated? what is the cutoff ratio?
6. For Figure 7A, representative 2DE image of U87-MG proteome for the three conditions should be labeled with different colors and overlapped together.
7. For Tables 2 and 4, font format was not uniform.
8. The label for y-axis in Figure 8 should be -log10(p-value).
9. For Figure 10, the illustration for the shapes (circle, triangle et al) was missing. For better reading of the figure, it would be better to label on the bank area around the dots.
10. For Figure 11, for the normalization of p-TFEB and p-mTOR, the loading control should be the expression level of TFEB and mTOR, respectively.
11. In Materials and Methods – Gene Expression Analysis, Qubit v.1 fluorometer plus Qubit RNA HS Assay Kit can be used to determine RNA concentration, but not for RNA purity.
12. In Line 152, 37◦C was in wrong format.
Comments on the Quality of English LanguageMinor editing of English language required
Round 2
Reviewer 1 Report
Comments and Suggestions for Authors
1、 Because both HMC3 and U87MG are glial cells, the experimental results from these two in vitro cell models alone are insufficient to establish a connection with neurodegenerative diseases and neuroprotective effects. Experiments with neuronal cells and related in vivo animal studies are essential.
2、 For the western blot in Figure 10, different groups should be run on the same gel and then transferred onto the membrane to maintain comparability.
Reviewer 3 Report
Comments and Suggestions for Authors
I agree with the revision
Author Response
Thank you for your positive response.